# Mapping of schistosomiasis and soil-transmitted helminthiases across 15 provinces of Angola

**Elsa Palma Mendes**[1], **Hajra Okhai**[2], **Rilda Epifânia Cristóvão**[1], **Maria Cecília Almeida**[1], **Nzuzi Katondi**[3], **Ricardo Thompson**[4], **Sylvain Mupoyi**[4], **Pauline Mwinzi**[4], **Onesime Ndayishimiye**[5], **Ferdinand Djerandouba**[5], **Mary Chimbilli**[5], **Julio Ramirez**[5], **Erna Van Goor**[5], **Sergio Lopes**[5]*

1 Neglected Tropical Disease Control Section, National Directorate for Public Health, Ministry of Health Angola, Luanda, Angola, 2 Institute for Global Health, University College London, London, United Kingdom, 3 World Health Organization Angola, Luanda, Angola, 4 Expanded Special Program for Elimination of Neglected Tropical Diseases, World Health Organization, Brazzaville, Republic of Congo, 5 The MENTOR Initiative, Huambo, Angola

* sergio@mentor-initiative.net

**Data Availability Statement:** Data cannot be shared publicly as it is MoH Angola data. Data are available from the MoH Institutional Data Access

## Abstract

### Introduction

Schistosomiasis (SCH) and soil transmitted helminthiases (STH) have been historically recognized as a major public health problem in Angola. However, lack of reliable, country wide prevalence data on these diseases has been a major hurdle to plan and implement programme actions to target these diseases. This study aimed to characterize SCH and STH prevalence and distribution in Angola.

### Methods

A country wide mapping was conducted in October 2018 (1 province) and from July to December 2019 (14 provinces) in school aged (SAC) children in 15 (of 18) provinces in Angola, using WHO protocols and procedures. A total of 640 schools and an average of 50 students per school (N = 31,938 children) were sampled. Stool and urine samples were collected and processed using the Kato-Katz method and Urine Filtration. Prevalence estimates for SCH and STH infections were calculated for each province and district with 95% confidence intervals. Factors associated with SCH and STH infection, respectively, were explored using multivariable logistic regression accounting for clustering by school.

### Results

Of the 131 districts surveyed, 112 (85.5%) are endemic for STH, 30 (22.9%) have a prevalence above 50%, 24 (18.3%) are at moderate risk (prevalence 20%-50%), and 58 (44.3%) are at low risk (<20% prevalence); similarly, 118 (90,1%) of surveyed districts are endemic for any SCH, 2 (1.5%) are at high risk (>50% prevalence), 59 (45.0%) are at moderate risk (10%-50% prevalence), and 57 (43.5%) are at low risk (<10% prevalence). There were

via geral@inis.ao or visit https://www.inis.ao/index.php/contactos.

**Funding:** This mapping was funded by ESPEN/WHO (https://espen.afro.who.int/). The study was also partly funded by The END Fund under the current NTD country support program under implementation by The MENTOR Initiative. The funders had no role in study design, data collection and analysis, decision to publish, or preparation of the manuscript.

**Competing interests:** The authors have declared that no competing interests exist.

higher STH infection rates in the northern provinces of Malanje and Lunda Norte, and higher SCH infection rates in the southern provinces of Benguela and Huila.

## Conclusions

This mapping exercise provides essential information to Ministry of Health in Angola to accurately plan and implement SCH and STH control activities in the upcoming years. Data also provides a useful baseline contribution for Angola to track its progress towards the 2030 NTD roadmap targets set by WHO.

### Author summary

Neglected Tropical Diseases (NTD) still affect nearly 1 billion people worldwide and are a major public health problem in Angola. Schistosomiasis (SCH) and soil transmitted Helminthiases (STH) affect disproportionally school aged children (SAC). In endemic areas, implementation of preventive chemoprevention through school-based Mass Drug Administration Campaigns is a key strategy used to reduce the burden of these infections. Mapping of schistosomiasis and soil transmitted helminthiases is essential to know where transmission occurs and is used to inform interventions planning. A country wide SCH and STH mapping was conducted across 15 of the 18 provinces of Angola. Parasitological analysis of nearly 32,000 children was conducted to detect SCH and STH infections and determine the prevalence of these diseases. Eighty Six percent of the mapped districts are endemic for STH and 22.9% have a prevalence above 50%. Similarly, 90% of surveyed districts are endemic for SCH. There were higher STH infection rates in the northern provinces of Malanje and Lunda Norte, and higher SCH infection rates in the southern provinces of Benguela and Huila. These results are of vital importance to map the prevalence of SCH and STH in Angola and to plan adequate interventions that support NTD control across the country.

## Introduction

Neglected Tropical Diseases (NTD) are a group of poverty-related diseases, which are often chronic conditions impact nearly 1 billion an individual's social and economic contributions worldwide [1]. Soil-transmitted helminthiases (STH) and schistosomiasis (SCH) can be controlled through interventions including preventive chemotherapy (often carried out as mass drug administration (MDA)) with impact and reinfection dependant on several factors such as the frequency, delivery and coverage of the campaigns, water and sanitation conditions, water contact patterns and sociodemographic factors [2–6].

Since the 2012 London Declaration on NTD, a global effort to eliminate NTDs was set and now reinforced with World Health Organization's (WHO) road map for neglected tropical diseases 2021–2030 [1]. Regular treatment, adequate monitoring and evaluation are deemed as critical to achieve 2030 targets. In this context, mapping of SCH and STH is essential to know where transmission occurs and inform targeted interventions [7].

In Angola, STH and SCH are a recognized public health problem [8], with approximately 5–10% of the population in need of preventive chemotherapy for both diseases [1]. There is little published information on the distribution of these diseases and existing data are either outdated or covering limited geographical areas [9,10]. Passive data collection carried out in the

mid 1900's showed increasing *Schistosoma haematobium* cases reported in a number of provinces [11,12] and a survey in Bié confirmed that more than half of the population had urogenital SCH [13]. Subsequent surveys confirmed the existence of high prevalence rates of *S. haematobium* in both Malange [14] and in Huila provinces [15–17]. Additional mapping work carried out in Benguela, and Luanda found 93% of SAC to be infected with *S. haematobium* [18]. From 1980's to 2000, the burden of SCH in Angola has been solely based on these estimations [19–21].

In contrast, for STH, surveys carried out in the 1950s in Zaire, Malange and Benguela estimated that up to 90% of the population was infected with *Hookworm* [11,14]. In Cuando Cubango, Huila and Kwanza Norte provinces, STH prevalence ranged from 65%-96% with *Hookworm* prevalence ranging from 40%-85% [15,22,23]. The first STH nationwide mapping conducted in 1964 registering prevalence around 75% in provinces like Huambo, Uige and Zaire [24]. Later surveys carried out in Bie found 86% of children had co-infections of 2 or more helminths [25]. While these referenced studies have shown a the presence of SCH and STH across Angola, their size and scope could not provide reliable prevalence estimates at district level, highlighting the need to conduct a country wide mapping.

In 2005, a STH and SCH mapping was carried out by the Angolan Ministry of Health (MoH) supported by the United Nations Children's Fund (UNICEF) and WHO based on ecological regions. Results showed estimated nationwide urogenital SCH prevalence in Angola was 28%, with higher incidence in the southern (40,6%), central plateau (39,6%) and northern provinces [26]. STH was reported at 40% prevalence across the country with *Ascaris* (25,0%) and *Hookworm* (9,8%) reported as dominant [26]. Data from northern and central Angola confirms the pattern of predominance of *Ascaris* in SAC but also highlights the burden of *Trichuris* infection [9,27,28].

In 2014, a baseline mapping of SCH and STH infections was conducted through a collaboration between the Angolan Ministry of Health (MoH) The MENTOR Initiative (MENTOR) and the End Fund in the provinces of Uige, Zaire and Huambo. The mapping measures SCH and STH prevalence at district level as recommended by WHO confirming the high burden of STH, particularly in Uige but also moderate prevalence of SCH infections (from 10%-15%) with significant variations between districts within each province [29].

Since 2014, preventive chemotherapy interventions have been implemented firstly in Huambo, Uige and Zaire provinces followed by Cuanza Sul, Bié and Cuando Cubango in 2017, and Bengo in 2019. Recognizing the need to arrange intervention based on reliable mapping data, the MoH, supported by World Health Organization, led a country wide SCH and STH mapping exercise in October 2018 (1 province) and from July to December 2019 (14 provinces). The objective of this study was to quantify the prevalence and distribution of these diseases across fifteen provinces in Angola in order to be able to adequately plan preventive chemotherapy interventions. A secondary aim of this project was to explore whether the presence of a latrine or water source in the school reduced the risk of infection.

## Methods

### Ethics statement

This mapping was approved by the Angola Ministry of Health Ethics Committee (Approval number 27/2018) in June 2018. Informed Consent was sought from participants parents. The team liaised with School directors prior to the survey to ensure parents were aware of the benefits and harms of participating in the survey. Parents were provided with an explanatory information sheet and a consent form to take home to decide. Parents were asked to send their children to school with the signed form if they agreed.

All students participated in the mapping voluntarily. Children were briefed on the objectives of the mapping and only took part if they verbally assented to participate.

All data was kept anonymous. No personal information was collected as children were identified through a unique identifier number. All children enrolled were treated with a dose of Albendazole and Praziquantel according to their height.

## Study design

A cross sectional survey using the standard WHO method for mapping [30] was implemented in October 2018 (Bengo province). Thereafter, from July to December 2019 in the remaining fourteen Angolan provinces not yet mapped for STH and SCH (Benguela, Bié, Cabinda, Cunene, Cuando Cubango, Cuanza Norte, Cuanza Sul, Lunda Norte, Lunda Sul, Malanje, Moxico, Huíla, Namibe and Luanda). A total of 131 districts were mapped. Parasitological examinations, knowledge attitudes and practices questionnaires were implemented in 640 public schools.

## School selection procedures

An average of five schools per district were selected but the number of schools selected by district was determined according to population data and the geographical area to be covered. The total number of schools per district ranged from 1 (in unpopulated areas and/or in highly concentrated urban areas) to 13 (in large geographical areas).

School selection was done in two stages: in the first instance, simple random sampling was conducted to select a defined number of schools per district. Then, the list was assessed with direction from local authorities to verify the geographical spread of schools selected across the district. Schools selected close to one another were purposively replaced by schools in locations known to be in areas of increased risk for SCH transmission. The proximity to fresh water sources (river, lakes, lagoons or swamp areas) were considered for this exercise to ensure over-sampling in these specific areas, as these provide the ideal ecological conditions for SCH transmission and are recommended as areas where mapping should be conducted[30].

## Study participants

Fifty children per school, 25 males and 25 females aged between 10–14 years old were invited to participate the day before the survey. School directors were asked to provide a list of all students to ensure systematic random selection.

For inclusion, only school-aged children resident in the study area for at least 2 years were considered to participate. Informed consent was requested of the child's parents the day before and only those carrying signed informed consent were included in the study.

Children who had taken any antiparasitic drug in the previous 6 months (particularly Albendazole, Mebendazole, Praziquantel, Ivermectin) were not included in the survey.

## Parasitological diagnosis

All children were provided with two plastic pots and requested to provide fresh stool and a urine sample. Kato Katz technique was used for analysing stool samples and urine filtration was used for the analysis of urine.

Kato Katz is a WHO reference technique for detecting and determining infection intensity for STH and *Schistosoma mansoni* allowing identification and quantification of these parasites [31] Microscopy using the Kato Katz technique requires fresh stool specimens, therefore analysis of specimens was conducted on site. The technique consisted of a microscopic examination

of a sample of stool to examine the number of eggs in the faeces. All samples were collected, processed, and examined on the same day. All eggs were counted within one hour of preparing the slides. A single slide per student was prepared and reading was done once in the day of the collection as recommended for operational mapping[30].

Urine filtration microscopy is the WHO standard technique for evaluating *Schistosoma haematobium* infection. A microscopic examination of a filter was used to collect the eggs of *S. haematobium* from 10 ml of urine. Macro-haematuria was visually inspected prior to the microscopic analysis of each sample.

### Data collection and management (ESPEN Collect)

Data was collected though standard questionnaires using the ESPEN collect tool. The tool was developed by the Expanded Special Project for the Elimination of Neglected Tropical diseases (ESPEN) to allow collection of survey data and inform school and administrative level prevalence in real time. The tool was shaped to integrate key parameters under assessment and adjusted to Angola's geographical regions by adding administrative boundaries to the mapping modules. ESPEN collect also allowed the generation of non-identifiable unique identifiers for every single child providing a useful resource to link parasitological data with school conditions and Knowledge, Attitudes and Practices (KAP) data.

The ESPEN Collect had four main questionnaires that were filled by a dedicated data manager in each team who was responsible to input all information of the survey.

Questionnaire 1. School Information sheet–This questionnaire collected information about school population (number of Students/teachers); Water source availability in the school and type of source to have water in the school; existence of freshwater bodies around the school; Presence and type of sanitation structure in the school; Presence and type of handwashing station in the school;

Questionnaire 2: Kato Katz sheet: This form provided collected individual data per student on the number of eggs counted of each species found in the slide (*S. mansosi*, *Ascaris Lumbricoides*, *Trichuris Thricuria*, *Hookworm* and others)

Questionnaire 3: Urine Filtration sheet: Also provided individual data per student about Macroscopic looking of the sample; Volume of Urine filtered and Number of *S. haematobium* *eggs*.

Questionnaire 4: Students' hygiene and risk behaviours: This questionnaire was used to all enrolled children and collected information about children gender, age, and hygiene behaviours such as usual place of defecation and freshwater bodies regular contact.

School location details were recorded in all questionnaires alongside. A single student identification code was generated for each participant that was used in all forms. This unique identified was used to merge the four datasets generated and analyse data gathered.

### Statistical analysis

Data were merged, cleaned, and analysed using Stata version 16 (College Station, TX: StataCorp LLC.). Prevalence (percentage and 95% confidence interval (CI)) and intensity of each infection (based on specified WHO thresholds[30]), any STH and any SCH infection were calculated based on the presence of eggs present in stool or urine samples, as appropriate, and presented by province and district (S1 and S2 Files).

STH and SCH risk for each district was determined using the calculated prevalence and based on the specified WHO thresholds [31]. The risk was mapped on a geographical map of Angola using ArcGIS version 10.3 (ESRI, Inc., Redlands, USA).

**Table 1. Geographical Distribution of sampled schools and children and key characteristics of the sample.**

| Province | Children | Schools | Male | Female |
|---|---|---|---|---|
| | 31938 | 640 | 15966 | 15972 |
| Bengo | 1549 (4.9%) | 31 (4.8%) | 780 (50.4%) | 769 (49.6%) |
| Benguela | 2150 (6.7%) | 43 (6.7%) | 1082 (50.3%) | 1068 (49.7%) |
| Bie | 2373 (7.4%) | 48 (7.5%) | 1192 (50.2%) | 1181 (49.8%) |
| Cabinda | 1150 (3.6%) | 23 (3.6%) | 575 (50.0%) | 575 (50.0%) |
| Cuando Cubango | 1850 (5.8%) | 37 (5.8%) | 925 (50.0%) | 925 (50.0%) |
| Cunene | 2100 (6.6%) | 42 (6.6%) | 1050 (50.0%) | 1050 (50.0%) |
| Huila | 3650 (11.4%) | 73 (11.4%) | 1826 (50.0%) | 1824 (50.0%) |
| Kwanza Norte | 2372 (7.4%) | 48 (7.5%) | 1195 (50.4%) | 1177 (49.6%) |
| Kwanza Sul | 2750 (8.6%) | 55 (8.6%) | 1377 (50.1%) | 1373 (49.9%) |
| Luanda | 1300 (4.1%) | 26 (4.1%) | 648 (49.8%) | 652 (50.2%) |
| Lunda Norte | 2450 (7.7%) | 49 (7.7%) | 1226 (50.0%) | 1224 (50.0%) |
| Lunda Sul | 1544 (4.8%) | 31 (4.8%) | 745 (48.3%) | 799 (51.7%) |
| Malanje | 2900 (9.1%) | 58 (9.1%) | 1449 (50.0%) | 1451 (50.0%) |
| Moxico | 2800 (8.8%) | 56 (8.8%) | 1396 (49.9%) | 1404 (50.1%) |
| Namibe | 1000 (3.1%) | 20 (3.1%) | 500 (50.0%) | 500 (50.0%) |

Finally, logistic regression using robust standard errors which accounted for clustering by school was used to explore whether the presence of a latrine or water source in the school was associated with STH or SCH infection, respectively. Models were adjusted for demographic factors including age, sex and province.

## Results

Over the survey period, 31,938 children were sampled from 640 schools across 131 districts in Angola and covered 15 of the 18 provinces across the country. Children were aged between 10 and 14 with a median age of 12 (interquartile range: 11–13) (Table 1).

### Prevalence of SCH infection

Overall, the sampled prevalence of any SCH infection amongst SAC children was 13.2% [95% CI: 12.8–13.5] with *S. haematobium* being the most prevalent species (12.6% [95% CI: 12.2–12.9]) compared to *S. mansoni* (0.9% [95% CI: 0.8–1.0]). The prevalence of any SCH infection was highest in Huila (32.3%), followed by Benguela (19.3%), Malanje (18.3%), and lowest in Lunda Sul (1.8%). Although *S. haematobium* was prevalent across all 15 sampled provinces, *S. mansoni* was only prevalent across 10 provinces. District specific prevalence with respective gender disaggregation can be found in S1 File.

When calculating risk of SCH infection, only two districts had a high risk (>50% prevalence) of SCH infection, 59 with a medium risk of SCH infection (10–50% prevalence) and 57 with a low risk of SCH infection (<10% prevalence). No SCH infections were reported from nine districts (Fig 1).

### Prevalence of STH infection

The prevalence of any STH infection was 24.1% [95% CI: 23.7–24.6] with *A. lumbricoides* being the most prevalent species (19.0% [95% CI: 18.6–19.5]) followed by Hookworms (5.8% [85% CI: 5.6–6.1]). *T. trichiura* (1.6% [95% CI: 1.4–1.7]) was the least prevalent species. The prevalence of any STH infection was highest in Kwanda Norte (69.7%), followed by Malanje

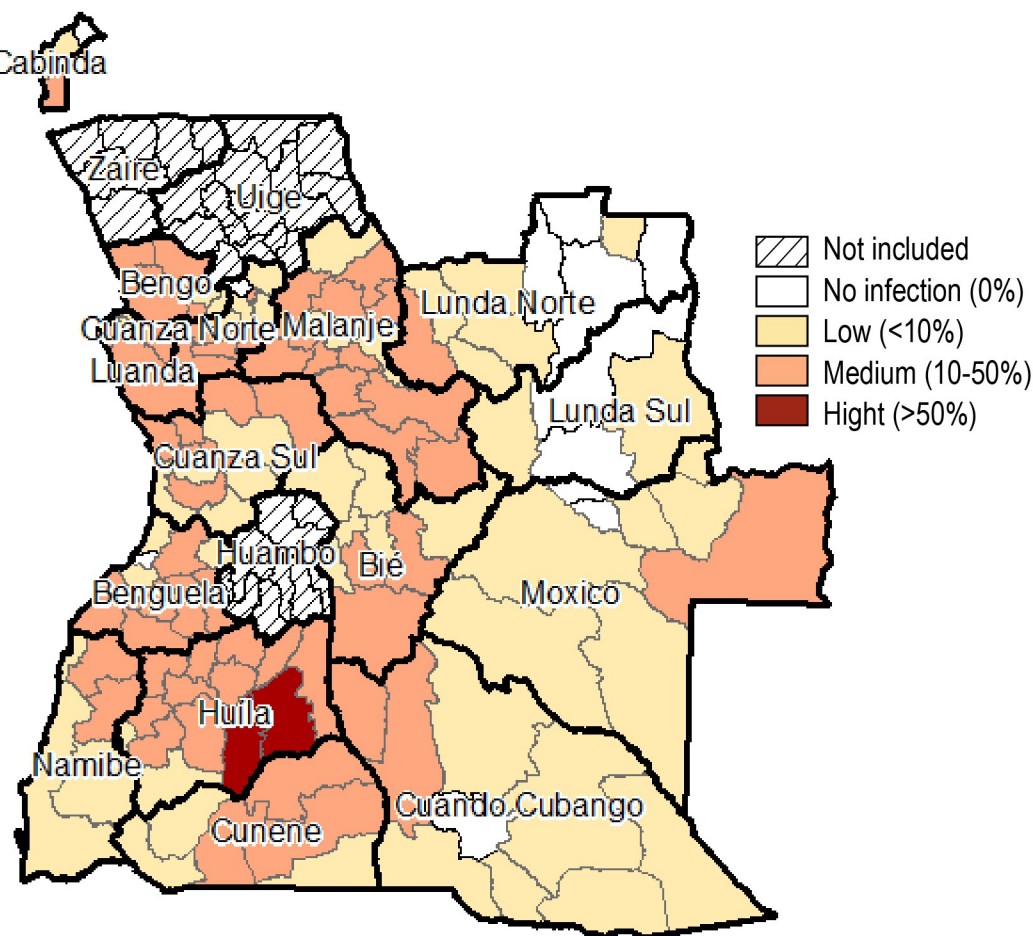

**Fig 1. Schistosomiasis risk (derived from Schistosomiasis prevalence) across 15 provinces of Angola (2018/2019).**
(Source: Ministry of Health Angola).

(55.9%), Lunda Norte (50.5%), and lowest in Namibe (0.8%). Five provinces (Bengo, Kwanza Norte, Lunda Norte, Lunda Sul and Malanje) had a prevalence of >20% (moderate risk) of *A. lumbricoides*. Only two provinces (Malanje and Moxico) had a moderate risk of Hookworm infection. All provinces had a low risk (<20% prevalence) of *T. trichiura*. District specific prevalence with respective gender disaggregation can be found in S2 File.

When calculating risk of STH infection by district, 30 districts had a high risk (>50% prevalence), 24 with a medium risk (20–50% prevalence) and 58 with a low risk (<20% prevalence) of STH infection. No STH infections were reported in 15 districts across the country (Fig 2).

## School data

School questionnaires were available for 639 schools (no data was recorded for one school in Namibe province), and therefore included in the following analyses. Only 30% (189/639) of schools reported having a water source in the school. For the majority of these schools, the water source was a protected fountain (26.5%; 50/189) or tap water (25.4%; 48/189) (S3 File).

Approximately 60% (385/639) of schools reported having latrine in the school grounds, with 37.7% (145/385) being a paved latrine, 26.8% (103/385) a non-paved latrine, 21.8% (84/385) a latrine with a flush and 12.7% (49/385) a ventilated improved pit (VIP) latrine.

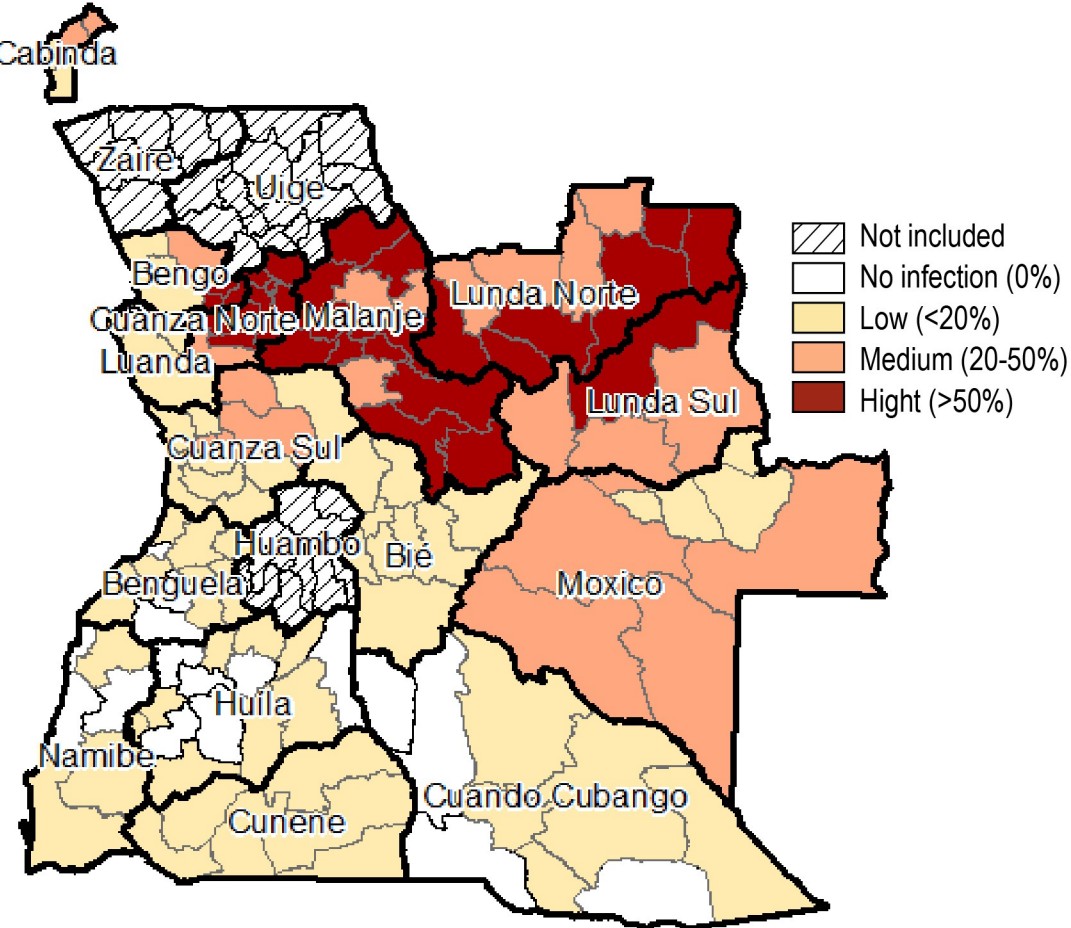

**Fig 2. Soil Transmitted Helminthiasis risk (derived from STH prevalence) across 15 provinces of Angola (2018/2019).** (Source: Ministry of Health Angola).

However, the majority (73.0%; 281/385) of schools with an available latrine reported never having water or toilet paper to use after using a latrine.

## Student behaviour questionnaire

Analysis showed a large proportion of missing data, discrepancies and conflicting answers. For that reason, these results are presented in S4 File but were not considered robust enough to be included in further analysis.

## Infection risk associated with latrine/water source in school

There was no association between the presence of a latrine (odds ratio (OR): 1.21 [95% confidence interval: 0.90, 1.62]) or water source (OR: 0.90 [0.67, 1.19]) in the school and SCH infection. This remained true in a multivariable logistic model (Table 2).

Although there seemed to be a reduced risk of STH infection with the presence of a latrine (OR: 0.69 [0.54, 0.89]) and water source (OR: 0.75 [0.57, 0.99]) in the school, these associations did not remain after adjustment for age, sex and province (Table 2).

**Table 2.** Logistic regression assessing the association between presence of latrine or water source in the school and SCH and/or STH infection adjusted for age, sex and province.

| | | Univariable | | Multivariable | |
|---|---|---|---|---|---|
| | | OR (95% CI) | p-value | OR (95% CI) | p-value |
| *SCH infection* | | | | | |
| Latrine in school | No | 1 | 0.21 | 1 | 0.72 |
| | Yes | 1.21 (0.90, 1.62) | | 1.04 (0.83, 1.31) | |
| Water source in school | No | 1 | 0.45 | 1 | 0.33 |
| | Yes | 0.90 (0.67, 1.19) | | 0.88 (0.68, 1.14) | |
| *STH infection* | | | | | |
| Latrine in school | No | 1 | 0.005 | 1 | 0.72 |
| | Yes | 0.69 (0.54, 0.89) | | 1.04 (0.83, 1.31) | |
| Water source in school | No | 1 | 0.05 | 1 | 0.33 |
| | Yes | 0.75 (0.57, 0.99) | | 0.88 (0.68, 1.14) | |

## Discussion

This SCH and STH mapping effort constitutes a landmark for NTD control in Angola. This has been the first country wide mapping exercise that sampled and collected SAC children data across several provinces to estimate the prevalence of these diseases. Despite the mapping exercises conducted in 2005 [26] and later in 2014 [29] and 2021 [9], this is the first mapping at country level that consistently follows WHO guidance for SCH and STH mapping in Africa [30].

Geographical distribution of SCH is consistent with historical data identifying high prevalence of this disease across Bié, Huila, Benguela, Bengo and Luanda provinces [9,10,13–18]. This is also in line with the 2005 mapping data that identified a higher prevalence of SCH across the central plateau [26]. *S. haematobium* prevalence is consistently higher than *S. mansoni* across these areas which is a consequence of the focal nature of SCH transmission, its association with human contact with infested water and the existence of a specific intermediate snail host [32].

Higher prevalence of STH was found in the northern provinces of Kwanza Norte, Malanje, Lunda Norte and Lunda Sul, with some schools mapped noting 100% of SAC infected with at least one STH. *Ascaris* has been the main infection found across these provinces, but *Hookworm* was frequently identified across Lunda Sul, Moxico and in Malanje, where there is a historical record of the disease [11,14]. These findings highlight the need to tailor communication interventions in these areas, particularly in impoverished rural areas where children tend to walk barefoot, a known risk factor for hookworm infection [33,34].

Less than a third of schools reported to have a water source in the school perimeter. Of these, only half had safe water source (protected fountain or tap water). Such results are in line with existing data about access to basic service water in Angola [35–37]. Similarly, sanitation information from schools mapped is aligned with existing information about sanitation access in schools in Angola [37]. These findings raise the need to improve water and sanitation conditions for SAC across the country. The low proportion of schools with sanitation equipment that had water or toilet paper to use after using a latrine suggests the need to invest in better sanitation equipment to ensure handwashing post defecation. Since poor WASH conditions are associated with increased risk of both SCH and STH transmission [38,39], it is essential to look at NTD control as an integrated approach that includes improvements of water and sanitation access and conditions in schools.

**Table 3. Number of districts in each province to be targeted for MDA according to mapped risk.**

| Província | N | SCH | | | STH | | |
|---|---|---|---|---|---|---|---|
| | | Low | Medium | High | Low | Medium | High |
| **Bengo** | 6 | 2 (33.3%) | 4 (66.7%) | - | 2 (33.3%) | 1 (16.7%) | 3 (50.0%) |
| **Benguela** | 10 | 3 (30.0%) | 6 (60.0%) | - | 7 (70.0%) | - | - |
| **Bie** | 9 | 6 (66.7%) | 3 (33.3%) | - | 9 (100.0%) | - | - |
| **Cabinda** | 4 | 2 (50.0%) | 1 (25.0%) | - | 2 (50.0%) | 2 (50.0%) | - |
| **Cuando Cubango** | 9 | 6 (66.7%) | 2 (22.2%) | - | 5 (55.6%) | - | - |
| **Cunene** | 6 | 3 (50.0%) | 3 (50.0%) | - | 6 (100.0%) | - | - |
| **Huila** | 14 | 2 (14.3%) | 10 (71.4%) | 2 (14.3%) | 8 (57.1%) | - | - |
| **Kwanza Norte** | 10 | 4 (40.0%) | 5 (50.0%) | - | - | 1 (10.0%) | 9 (90.0%) |
| **Kwanza Sul** | 12 | 5 (41.7%) | 7 (58.3%) | - | 8 (66.7%) | 4 (33.3%) | - |
| **Luanda** | 9 | 5 (55.6%) | 3 (33.3%) | - | 4 (44.4%) | 1 (11.1%) | - |
| **Lunda Norte** | 10 | 5 (50.0%) | 1 (10.0%) | - | - | 4 (40.0%) | 6 (60.0%) |
| **Lunda Sul** | 4 | 2 (50.0%) | - | - | - | 3 (75.0%) | 1 (25.0%) |
| **Malanje** | 14 | 3 (21.4%) | 11 (78.6%) | - | - | 3 (21.4%) | 11 (78.6%) |
| **Moxico** | 9 | 6 (66.7%) | 1 (11.1%) | - | 4 (44.4%) | 5 (55.6%) | - |
| **Namibe** | 5 | 3 (60.0%) | 2 (40.0%) | - | 3 (60.0%) | - | - |

[40][41–44]When controlling for age, sex and province, the presence of water and latrine in school was not associated with STH nor with SCH infections. This may be related to the absence of water and sanitation conditions in communities where children live. Children may have access to these conditions in schools but have limited or no access to adequate water and sanitation at home, a problem that has been previously identified [40]. Another major contributor may be the limited use of existing sanitation structures in schools [37]. Unfortunately, due to the high proportion of missing behavioural data, it was not possible to accurately assess these. But, when looking at the proportion of children reporting to urinate or defecate in school latrines, approximately only a fifth of children report to do so. WASH in school is essential to provide essential infrastructure, to foster its use and the adoption of healthy sanitation behaviours. This is not new as the requirement for investments in WASH in schools in Angola have been raised historically by UNICEF [37].

Results obtained by this mapping are essential to adequately plan MDA campaigns and ensure geographical areas are targeted based on the need. The Angola NTD Masterplan 2021–2025 [41] (final version awaiting ministerial approval) integrates this mapping data and projects its long-term results in line with WHO roadmap [1]. However, the main identified risk to attain those results is linked to the chronical lack of support to NTD activities in Angola. So far, financial support to conduct MDA has been provided by WHO and The MENTOR Initiative (through an End Fund Grant). MDA activities have been implemented across seven provinces with gradual appropriation of activities from several district and provincial health and education authorities over time. Nevertheless, mapping results "demand" the rapid scale up of MDA to several districts. For STH, 30 districts should be targeted bi-annually and 24 targeted annually corresponding to an estimated total of 2.650.000 SAC treated annually. For SCH, 2 districts should be targeted annually and 65 targeted at least once in 5 years, in an estimated total of 2.950.000 SAC children to be treated (Table 3). These treatment efforts need to be integrated with community level MDA, particularly where overlap of treatment with Albendazole is foreseen. However, as lymphatic filariasis mapping data is outdated, it is hard to accurately integrated these strategies.

One of the major limitations of this study is related to the sampling procedures and its impact on calculating SCH prevalence at sub district level. Sampling procedures were used to estimate prevalence at district level. Due to the focal nature of SCH, WHO recommends sub district estimates for SCH prevalence [42] and produced a tool to extrapolate prevalence at the sub-district level out of district level results. Such tool has also been adopted in Angola to refine mapping results. Another operational limitation of this was the non-inclusion of three provinces: Huambo, Uige and Zaire. These provinces have been mapped in 2014 and MDA have been implemented since. Considering that mapping should occur after 5 years of MDA, it was agreed that an impact assessment would take place in 2020 (delayed to 2021 due to the COVID-19 pandemic).

The use of single stool and urine samples for estimating the prevalence is also a major limitation to interpret the findings. The Kato Katz technique has recognized accuracy limitations particularly for low intensity infections. Therefore, it would have been preferable to collect several samples over time to improve accuracy [43]. However, considering the scope and size of the mapping needed to characterize STH and SCH prevalence in Angola and the available resources, such methods would have been unfeasible to implement across the country.

The inability to explore individual hygiene behaviours constitute a major challenge in order to fully understand STH and SCH risk behaviours of SAC in Angola. Nevertheless, this descriptive analysis, exploring the presence of a latrine or water source in the school provides important insights about key regions that have the highest risk of contracting SCH and STH. These should be targeted through information campaigns aiming to reduce well identified risk behaviours such as poor handwashing practices, open defecation, walking barefoot and bathing in rivers, dams, or lakes.

Since egg counting procedures and recording were not consistently done across all provinces, it was not possible to compute infection intensities. This poses a limitation to follow up the impact of NTD control interventions on infection intensity over time. In addition, it was not possible to map pre-SAC or other risk groups which constitutes a limitation to understand better transmission patterns across the whole Angolan population, including the role of adults in transmission of STH and ACH. Previous studies identified pregnant women to be at risk of infection in Angola [9]. Future research in Angola should consider the inclusion of adults that are not generally targeted in SCH and STH control interventions.

## Conclusions

This first ever STH and SCH mapping in Angola achieved it main objective of quantifying the prevalence and distribution of these infections across the country. Results are of vital importance to map the prevalence and geographical distribution of these diseases and plan adequate interventions that support NTD control in Angola and contribute to WHO 2030 defined NTD control targets. Water and sanitation conditions in schools across Angola are still scarce and may be a significant factor contributing for the high endemicity of some NTD in Angola.

## Supporting information

**S1 File. Schistosomiasis haematobium, Schistosomiasis mansoni and Any Schistosomiasis prevalence (and 95% CI) by province and district with gender disaggregation.**
(XLSX)

**S2 File. Ascaris Lumbricoides, Trichuris, Hookworm and any Soil Transmitted Helminth prevalence (and 95% CI) by province and district with gender disaggregation.**
(XLSX)

**S3 File. WASH conditions in schools sampled.**
(XLSX)

**S4 File. Individual students practices related to WASH.**
(XLSX)

## Acknowledgments

The authors would like to thank all lab workers and data managers who significantly contributed to ensure the implementation of field work. Likewise, to all logistic staff involved (drivers, mobilizers) who supported daily field activities. The authors also acknowledge all the Provincial and Municipal authorities support to the implementation of this mapping and, to Education and Health departments at provincial and district level who facilitated all contacts and, in the vast majority of the cases provided housing for field teams in remote settings. A major appreciation should also be given to all School Directors and teachers that contributed in a very positive way to make this mapping happen as they played a major role in mobilizing parents and children to enrol in the mapping. Finally, the authors would like to thank all parents and children engaged in the mapping for their vital contribution to better understand neglected tropical diseases in Angola.

## Author Contributions

**Conceptualization:** Elsa Palma Mendes, Ricardo Thompson, Sylvain Mupoyi, Onesime Ndayishimiye.

**Data curation:** Hajra Okhai, Rilda Epifânia Cristóvão, Mary Chimbilli, Julio Ramirez, Erna Van Goor, Sergio Lopes.

**Formal analysis:** Hajra Okhai, Erna Van Goor, Sergio Lopes.

**Funding acquisition:** Nzuzi Katondi, Sylvain Mupoyi.

**Investigation:** Elsa Palma Mendes, Ricardo Thompson, Sylvain Mupoyi, Onesime Ndayishimiye.

**Methodology:** Elsa Palma Mendes, Nzuzi Katondi, Ricardo Thompson, Sylvain Mupoyi, Pauline Mwinzi, Onesime Ndayishimiye.

**Project administration:** Elsa Palma Mendes, Rilda Epifânia Cristóvão, Maria Cecília Almeida, Nzuzi Katondi, Ferdinand Djerandouba.

**Resources:** Sylvain Mupoyi, Mary Chimbilli.

**Supervision:** Elsa Palma Mendes, Rilda Epifânia Cristóvão, Nzuzi Katondi, Sylvain Mupoyi, Onesime Ndayishimiye, Ferdinand Djerandouba, Mary Chimbilli, Julio Ramirez, Erna Van Goor.

**Validation:** Nzuzi Katondi, Onesime Ndayishimiye, Mary Chimbilli, Julio Ramirez, Erna Van Goor.

**Writing – original draft:** Hajra Okhai, Sergio Lopes.

**Writing – review & editing:** Elsa Palma Mendes, Hajra Okhai, Rilda Epifânia Cristóvão, Maria Cecília Almeida, Nzuzi Katondi, Sylvain Mupoyi, Pauline Mwinzi, Ferdinand Djerandouba, Mary Chimbilli, Julio Ramirez, Erna Van Goor, Sergio Lopes.

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
