## [Editor Report · Decision Letter 0]

9 Aug 2021

Dear Mr Lopes,

Thank you very much for submitting your manuscript "Mapping of Schistosomiasis and Soil-Transmitted Helminths across 15 provinces of Angola" for consideration at PLOS Neglected Tropical Diseases. 

When conducting the initial review of your submission, I noticed that there were a number of indications that the submitted version of the document was not the final version. For example, the submitted version has at least one reference place holder (i.e., [ref]), questions from co-authors left in the text associated with Table 1, supplementary materials referenced in the text as Appendix x, inconsistent use of abbreviations, etc. I would ask that you please address these issues and submit a revised version of your paper. Once an updated version has been submitted, the paper will be considered for review. Please note that this is not a guarantee that your manuscript will be accepted for publication.

Sincerely,

Christine M. Budke

Deputy Editor

When conducting the initial review of your submission, I noticed that there were a number of indications that the submitted version of the document was not the final version. For example, the submitted version has at least one reference place holder (i.e., [ref]), questions from co-authors left in the text associated with Table 1, supplementary materials referenced in the text as Appendix x, inconsistent use of abbreviations, etc. I would ask that you please address these issues and submit a revised version of your paper. Once an updated version has been submitted, the paper will be considered for review. Please note that this is not a guarantee that your manuscript will be accepted for publication.

Figure Files:

Data Requirements:

Reproducibility:

References

---

## [Decision Letter · Decision Letter 1]

25 Nov 2021

Dear Mr Lopes,

Thank you very much for submitting your manuscript "Mapping of Schistosomiasis and Soil-Transmitted Helminths across 15 provinces of Angola" for consideration at PLOS Neglected Tropical Diseases. As with all papers reviewed by the journal, your manuscript was reviewed by members of the editorial board and by several independent reviewers. In light of the reviews (below this email), we would like to invite the resubmission of a significantly-revised version that takes into account the reviewers' comments. 

We cannot make any decision about publication until we have seen the revised manuscript and your response to the reviewers' comments. Your revised manuscript is also likely to be sent to reviewers for further evaluation.

Sincerely,

Christine M. Budke

Deputy Editor

Reviewer's Responses to Questions

**Key Review Criteria Required for Acceptance?**

**Methods**

-Are the objectives of the study clearly articulated with a clear testable hypothesis stated?

-Is the study design appropriate to address the stated objectives?

-Is the population clearly described and appropriate for the hypothesis being tested?

-Is the sample size sufficient to ensure adequate power to address the hypothesis being tested?

-Were correct statistical analysis used to support conclusions?

-Are there concerns about ethical or regulatory requirements being met?

Reviewer #1: Yes

Reviewer #2: 

The data availability statement is incorrect because when it says where can the data be found, the authors have put these data cannot be shared. 

Line 11 of the abstract says 31,938 children in the study but in line 10 of the author summary says 13.000 which I presume is supposed to be 13,000 but is also inconsistent with the abstract or unclear if the correct number – what is the almost 32,000 if not the number of children included? In future do not restart page numbers for each section of the document. It makes referencing the correct part difficult, especially seeing as you have no page numbers.

Introduction Line 7 – it is also to do with water contact patterns, sociodemographic factors etc, these all contribute to the force of infection. 

Line 13 – you have referred to these as abbreviations above so SCH and STH should be abbreviations consistently. Also, schistosomiasis is the disease so does not need to be capitalised, but Schistosoma haematobium on line 19 has not been taxonomically declared before and should in this instance be spelt out with a capitalised S. 

Lines 24-25 – What do you mean by this sentence? All prevalence/ burden measures in any location are estimates as nowhere is the whole community sampled. 

Line 7 page 2 of your introduction – is MENTOR an acronym? 

The introduction would benefit from reference to the maps. It is quite listy in terms of locations and previous mapping efforts. Perhaps in the introduction, some guidance from the authors for the readers as to whether the overall prevalence and intensities of infections for schistosomiasis or STH are going up or down? Why is this mapping effort so important? As it stands, it seems like there have been regular and ongoing efforts to map these diseases. 

Line 20 School selection criteria – close not closed? 

Line 24 School selection criteria – some clarification for why you wanted to oversample in these locations would be beneficial. 

Line 10 Parasitological diagnosis – mansoni should not be capitalised. It has also not been taxonomically described yet in the text and should be fully spelt out in this instance. 

Line 15 Parasitological diagnosis – this is not true. The Kato-Katz method lacks sensitivity, egg excretion from the human host is highly over dispersed and does not necessarily reflect the density of female worms – there are a few hypotheses for this including density dependent reproduction, host acquired immunity and worm senescence. As it is essentially incorrect, please remove this sentence. Further to this, you do not explain at all how many stool samples you use, or over how many days (Lamberton, 2014). The methods generally lack any useful detail and give too much opinion. 

Lines 24-33 are superfluous. If you really want to include them put them in an appendix but for the purpose of the main text I would remove them. 

Lines 1-9 ESPEN data collection – Rather than describing what the ESPEN tool is, I would like to understand clearly and concisely what it was used to collect. This is not clear from the text. This can be significantly streamlined. 

Line 21 Statistical Analysis – you have a (ref) that I presume should be a reference. 

Line 30 Statistical Analysis – when you say clustering here, do you mean that the school was considered as a random effect in the model? Does this mean you did no analysis on infection intensity despite having all of these egg counts? I think this is a really important point. You could have high prevalence of low intensity infections. Furthermore, we know that the KK method performs poorly in low prevalence and low infection intensity settings – without an analysis on the intensities how can we really tell if this mapping is reflective of true infection distribution? I’d be likely to think there are more infections than this has detected if the intensities are low and maybe more likely to think it has done a good job at characterising infections if the counts are very high. With such huge sample sizes I am also unsure why you needed to do a univariate analysis first. This method notoriously produces spurious results. Did you not include any interactions or random effects either? I feel the statistical analysis here is either lacking effort and rigour or maybe lacking a more substantial technical description? I think I will not understand which it is until at least a more detailed technical description of the analysis is provided. Indeed, if this is such an important mapping effort, then peoples quality of life can depend on these results meaning statistical rigour is essential. Also, I wonder what the malaria treatment situation is given the prevalence of malaria in Angola and that malaria treatment has anti-schistosomal effects. “Data” is a plural word so reporting should be “no data were” or “not data are” rather than “no data was” and “no data is”. 

Discussion line 21 – do you think communication is needed for impoverished children to wear shoes or do you think this is a funding issue where funds should be diverted to enable shoe wearing? I think you really need to discuss the fact that egg-based diagnostics, particularly for SCH, is simply not an adequate tool for capturing true prevalence and intensity measures. This is not mentioned at all in your discussion and is a major problem. Given that there is no description for how many counts per stool over how many days, I am presuming also that just one count was done. This is also inadequate and grossly underestimates prevalence and infection intensity. These are major limiting factors to the credibility of the estimates provided. Also, in the context of the new WHO roadmap, what is the part played by animals in the maintenance of S. haem? Given recent work that has been published (Borlase, 2021 Leger, 2020) this deserves a comment in the discussion. You also have no comment of the fact that you only sampled children. Adults can harbour incredible infections and manifest chronic morbidity more often. In such instances then, who is really suffering the most from infection, children with high egg shedding rates, or adults with low egg shedding rates but high levels of morbidity (caused by the eggs not shed). you say that the WHO give guidance for taking district level estimates and turning them into subdistrict estimates. What is this method?

Reviewer #3: Authors made a good attempt in investigating a very important issue on evidence guided decision making. however, further clarity on the how the results reflect on the objective is required. In terms of the results, authors needs to bring out variate analysis output gender and age variations especially at province levels. 

The sample size is sufficient enough for this types of surveys an perceived adequate to prove useful information.

No ethical issues of significance were noted, the authors however needs to demonstrate coherently how consent have been given, written, or signed, could the parents read, or who was available to explain to parents to seek consent, does consent seeking involve translations. was there any participants excluded for reasons of non consent. 

The objective of this survey, which is stated as "mapping the epidemiology of SCH and STH" should be defined, to give give a grasp of what the authors meant by mapping the epidemiology of SCH and STH.

**Results**

-Does the analysis presented match the analysis plan?

-Are the results clearly and completely presented?

-Are the figures (Tables, Images) of sufficient quality for clarity?

Reviewer #1: Yes

Reviewer #2: (No Response)

Reviewer #3: Table 1, 2 and 3, need improvement on the appropriate titles, and content. Table 1 could incorporate more information on demographics of the study participants at province or municipal levels. 

The presentation of the results and the discussion should relate to the objective including how it will guide treatment arrangements.

How does the integrated treatment relate to or affect the wider community based MDA is not very clear.

**Conclusions**

-Are the conclusions supported by the data presented?

-Are the limitations of analysis clearly described?

-Do the authors discuss how these data can be helpful to advance our understanding of the topic under study?

-Is public health relevance addressed?

Reviewer #1: Yes

Reviewer #2: (No Response)

Reviewer #3: Authors should be concise but conclusive on the final take on the extent in which the objectives of the study had been achieved and how it will be used to guide strategic prevention and control strategies.

The public health relevance of the study had been achieved in practice. Although the information presented should be more coherent in both writeup style, arrangement and flow of the information being presented and expressions.

**Editorial and Data Presentation Modifications?**

Reviewer #1: • Page 9, line 3 – “individuals’” should be “individual’s”

• Throughout: “S. Haematobium” should be “S. haematobium”. Same with S. Mansoni.

• Throughout: “Schistosomiasis” should be “schistosomiasis” unless at the start of a sentence.

• Page 9, line 35, recommend changing “global” to “nationwide”.

• Page 10, line 4: “remainder” should probably be “remaining”

• Page 10, line 7: is “municipalities” here the same as “districts”?

• Page 14, lines 5-6: “Informed Consent forms were sought near participants parents” should probably be “Informed Consent forms were sought from participants’ parents”.

• Page 14, line 6: “Team elements” might be better as “Team members” or “Team leaders”.

• Page 16, lines 21 and 22: “pavement” might be better as “paved”.

• Page 20, line 34: “age’ could be removed.

• Page 21, line 3: “prevalence sub-district level” should probably be “prevalence at the sub-district level”

Reviewer #2: (No Response)

Reviewer #3: Line numbering was set by the authors to be repetitive per page and made review especially making reference to specific lines very cumbersome. Authors should set line numbers to be continuous instead.

**Summary and General Comments**

Reviewer #1: This is a well written and very interesting paper looking at the distribution of SCH and STH in Angola. The sample size is extensive – stretching to 640 schools, 32,000 children across 131 districts. These data will clearly be of use in planning SCH and STH interventions in the country and as a staging post to 2030 targets.

A few questions outlined below, most are minor.

Comments

• Abstract – results. When saying infection was related to age and sex, I’d recommend making clear in what direction. E.g. older people more likely to be infected. Males more likely to be infected. This is included in the results section, add to abstract.

• Abstract – mention whether you are referring to S.mansoni or S.haematobium.

• Author summary – this mentioned a sample size of 13,000 children, but 32,000 are referenced in the abstract. Which is accurate?

• Methods: For children indicating that they had taken parasitic drugs in the last six months, how many were there?

• Methods: “Schools selected that were closed to each other were purposively replaced by schools in locations known to be in areas of increased risk for Schistosomiasis transmission.” I think that’s a fine way to proceed, but we should be aware that it doesn’t give rise to unbiased estimates of prevalence.

• Methods: Student behavior questionnaire: agree that if data are not robust enough, they should not be used for analysis. What is the reason for that – due to students not understanding the questions? Methodological flaws?

• Methods: For age, how is the odds ratio calculated. For each additional year of age? Or comparing age categories?

• Page 16, lines 29-31. Interesting that children were less likely to have SCH infection if they had STH infection (but not in the multivariate analysis). Why do the authors think that is? 

• Page 20, lines 17-19: “Overall, data indicate that WASH in school investments are essential to provide essential infrastructure and to foster its use and the adoption of healthy sanitation behaviours”. I agree that there are so many reasons why school WASH infrastructure should be improved. But this paper does not show SCH / STH infection to be one of them.

• Could consider bringing Additional File 5 (number of districts in each province to be targeted for MDA) into main body. But not compulsory.

Reviewer #2: (No Response)

Reviewer #3: This survey is an important activity as part of the NTD elimination roadmap, and information generated will be very useful in guiding efforts in intervention. What the authors need is review the writing style, grammar, expression, flow and other typographical errors. Analysis and results should be more extensive in portraying province and municipality variance for ages and sexes and the strategy for treatment and other intervention like WASH, on the basis of evidence of level of prevalence high, medium and low.

PLOS authors have the option to publish the peer review history of their article (what does this mean?). If published, this will include your full peer review and any attached files.

Reviewer #1: Yes: Michael French

Reviewer #2: No

Reviewer #3: Yes: Yaya Camara
---

## [Decision Letter · Decision Letter 2]

10 Apr 2022

Dear Mr Lopes,

Thank you very much for submitting your manuscript "Mapping of Schistosomiasis and Soil-Transmitted Helminths across 15 provinces of Angola" for consideration at PLOS Neglected Tropical Diseases. As with all papers reviewed by the journal, your manuscript was reviewed by members of the editorial board and by several independent reviewers. The reviewers appreciated the attention to an important topic. Based on the reviews, we are likely to accept this manuscript for publication, providing that you modify the manuscript according to the review recommendations. 

Sincerely,

Christine M. Budke

Deputy Editor

Reviewer's Responses to Questions

**Key Review Criteria Required for Acceptance?**

**Methods**

-Are the objectives of the study clearly articulated with a clear testable hypothesis stated?

-Is the study design appropriate to address the stated objectives?

-Is the population clearly described and appropriate for the hypothesis being tested?

-Is the sample size sufficient to ensure adequate power to address the hypothesis being tested?

-Were correct statistical analysis used to support conclusions?

-Are there concerns about ethical or regulatory requirements being met?

Reviewer #2: (No Response)

Reviewer #3: I am of the opinion the the authors should limit the objective to the distribution of the said diseases, something like the prevalence and distribution. The term burden seem more complex to be established and seems inadequately addressed by the data presented.

**Results**

-Does the analysis presented match the analysis plan?

-Are the results clearly and completely presented?

-Are the figures (Tables, Images) of sufficient quality for clarity?

Reviewer #2: (No Response)

Reviewer #3: (No Response)

**Conclusions**

-Are the conclusions supported by the data presented?

-Are the limitations of analysis clearly described?

-Do the authors discuss how these data can be helpful to advance our understanding of the topic under study?

-Is public health relevance addressed?

Reviewer #2: (No Response)

Reviewer #3: (No Response)

**Editorial and Data Presentation Modifications?**

Reviewer #2: (No Response)

Reviewer #3: (No Response)

**Summary and General Comments**

Reviewer #2: This is the second time I have handled this manuscript. Given the satisfactory response to the original review I just have a few additional minor things: 

First, I totally appreciate the amount of effort that has gone into this mapping. My commentary on the lack of sensitivity with a single Kato-Katz was not a slight on your effort but on the continued recommendation from large bodies such as WHO, for its use, and thus a continued lack of financial support provisioned for improved diagnostics and labour. The 2030 goal is ambitious and if we cannot truly understand the dynamics how will it be possible to reach. As such, I hoped to see in the original version, a more thorough discussion of this as this is where experts such as yourselves can have your say and highlight this dire need. Indeed you have responded to my request for more of this in the discussion with a small section but I stand by my original review comment - that there are additional complexities to control that the WHO mapping protocol just does not capture. I do not expect you to add more to the discussion at this point, the manuscript is fine as it is, but just food for thought. 

Line 17 Parasitological diagnosis - remove the fullstop after Schistosoma 

Line 23 ESPEN collect section - Expanded not Espanded

Reviewer #3: (No Response)

PLOS authors have the option to publish the peer review history of their article (what does this mean?). If published, this will include your full peer review and any attached files.

Reviewer #2: No

Reviewer #3: Yes: Yaya Camara

Figure Files:

Data Requirements:

Reproducibility:

References

---

## [Editor Report · Decision Letter 3]

1 May 2022

Dear Mr Lopes,

We are pleased to inform you that your manuscript 'Mapping of schistosomiasis and soil-transmitted helminthiases across 15 provinces of Angola' has been provisionally accepted for publication in PLOS Neglected Tropical Diseases.

Best regards,

Christine M. Budke

Deputy Editor

Christine Budke

Deputy Editor

---

## [Editor Report · Acceptance letter]

20 Jun 2022

Dear Mr Lopes,

We are delighted to inform you that your manuscript, "Mapping of schistosomiasis and soil-transmitted helminthiases across 15 provinces of Angola," has been formally accepted for publication in PLOS Neglected Tropical Diseases.

Best regards,

Shaden Kamhawi

co-Editor-in-Chief

Paul Brindley

co-Editor-in-Chief
